# Multispecies Bacterial Bio-Input: Tracking and Plant-Growth-Promoting Effect on Lettuce var. *sagess*

**DOI:** 10.3390/plants12040736

**Published:** 2023-02-07

**Authors:** Santiago A. Vio, María Lina Galar, María Cecilia Gortari, Pedro Balatti, Mariana Garbi, Aníbal Roberto Lodeiro, María Flavia Luna

**Affiliations:** 1Centro de Investigación y Desarrollo en Fermentaciones Industriales, CINDEFI (CONICET/UNLP), Calle 50 227, La Plata 1900, Argentina; 2Comisión de Investigaciones Científicas de la Provincia de Buenos Aires (CIC-PBA), Calle 526 e/ Calles 10 y 11, La Plata 1900, Argentina; 3Centro de Investigaciones de Fitopatología, CIDEFI (CIC–UNLP), Calle 60 y 119, La Plata 1900, Argentina; 4Climatología y Fenología Agrícola, Facultad de Ciencias Agrarias y Forestales, UNLP, Calle 60 y 119, La Plata 1900, Argentina; 5Instituto de Biotecnología y Biología Molecular, IBBM (CONICET/UNLP), Calle 47 y 115, La Plata 1900, Argentina; 6Genética, Facultad de Ciencias Agrarias y Forestales, UNLP, Calle 60 y 119, La Plata 1900, Argentina

**Keywords:** co-inoculation, colonization, lettuce, multispecies bio-input, plant-growth-promoting bacteria

## Abstract

The use of multispecies bacterial bio-inputs is a promising strategy for sustainable crop production over the use of single-species inoculants. Studies of the use of multispecies bio-inputs in horticultural crops are scarce, not only on the growth-promoting effects of each bacterium within the formulation, but also on their compatibility and persistence in the root environment. In this work, we described that a multispecies bacterial bio-input made up of *Azospirillum argentinense* Az39, *Gluconacetobacter diazotrophicus* PAL-5, *Pseudomonas protegens* Pf-5 and *Bacillus* sp. Dm-B10 improved lettuce plant growth more effectively than when these strains were inoculated as single-species bio-inputs. Bacteria persisted together (were compatible) and also colonized seedling roots of lettuce plants grown in controlled conditions. Interestingly, colonization was highly related to an early and enhanced growth of seedlings grown in the nursery. A similar effect on plant growth was found in lettuce plants in a commercial greenhouse production in the peri-urban area of La Plata City, Buenos Aires, Argentina. To our knowledge, this is the first study demonstrating that a synthetic mixture of bacteria can colonize and persist on lettuce plants, and also showing their synergistic beneficial effect both in the nursery greenhouse as well as the commercial production farm.

## 1. Introduction

Plants are naturally colonized by bacterial communities located in the rhizosphere, the phyllosphere and even the endosphere [1]. Many of these bacteria play crucial roles in plant health and growth, and are commonly called Plant-Growth-Promoting Bacteria (PGPB) [2]. These bacteria have the ability to improve plant growth through different processes that include nitrogen fixation, phytohormone production and phosphate solubilization, and they also might induce components of the plant immune system that help plants to cope with abiotic and biotic stresses, among other factors. Consequently, a large variety of bacteria has been isolated from different plant environments and evaluated with in vitro and in vivo assays to determine their efficacy as PGPB on different agricultural crops [3,4]. Among bacteria, several genera, such as *Bradyrhizobium*, *Azospirillum*, *Bacillus*, *Gluconacetobacter* and *Pseudomonas*, have been characterized and used as PGPB worldwide [5]. Some PGPB strains of these genera have been used to formulate commercial single-species bio-inputs currently used as biofertilizers, biostimulants or biocontrol agents to promote crop growth and production as well as control disease [3,6,7]. However, in the last decade, co-inoculation or inoculation with multiple PGPB strains from either the same or different taxonomical groups—multispecies inoculants or second-generation inoculants—have acquired an important place not only in the topics of scientific research but also in the bio-input industry [8,9,10]. The design and formulation of multispecies bio-inputs might be advantageous over first-generation bio-inputs (single-strain), because: (1) microorganisms most probably act in a cooperative manner by providing nutrients, removing inhibitory products, and stimulating beneficial physiological traits; (2) such a diverse array of organisms might improve adaptation to a wide range of environmental conditions; (3) provision of more than one PGPB with equal or different modes of action might improve the ability of a bacterial community to promote plant growth [9]. In this way, several PGPB co-inoculation studies with promising results have been reported in crops such as soybean, maize, potato and wheat, among others [11,12,13,14,15,16]; however, there are few reports regarding the use of such tools on studies on horticultural crops [9,17,18]. Also, despite the enormous potential of microorganisms to promote plant growth either alone or in a mixture, the reproducibility of their beneficial effects in the field varied too much, which is frequently related to a failure of the inoculated microorganisms to colonize and establish within new environments [4,16]. Therefore, for a successful use of these potentially interesting new technologies of synthetic mixtures of bacterial, after selecting microorganisms according to their mode of action aimed toward multifactorial benefits, it is necessary to ensure not only the coexistence of the different microorganisms but also their ability to colonize the plants organs once they were inoculated [8,16].

Lettuce (*Lactuca sativa* L.) is one of the most highly consumed fresh raw vegetables in the world [19]. In Argentina, considering production and consumption, it is the most important leaf crop and the third-most important horticultural crop after potato and tomato [20]. It is cultivated throughout the country, mainly under cover and around large urban centers, in areas known as “green belt vegetables”, highlighting the La Plata Horticultural Belt (CHP) as one of the most important in amount and value of the production [20]. As it occurs with other vegetables [21], the lettuce production is divided into two stages: (1) a seedling stage in trays from sowing to transplanting, performed in nurseries in a greenhouse, and (2) a productive stage, from transplant to harvest, performed in a producer farm, greenhouse or field. This intensive system of lettuce production has a high demand for nutrients in the form of fertilizers and agrochemicals to avoid pests and diseases in both crop stages [22]. It has been reported that lettuce crops show a favorable response after inoculation with a single or a mixture of PGPB of different genera such as *Azospirillum*, *Azotobacter*, *Pseudomonas*, *Rhizobium* and *Bacillus*, among others [18,23,24,25,26,27], even in combination with fungi [28]. Such treatments led to larger plants with more biomass, but other aspects also were affected, such as the germination rate. Argentina is a country with early adoption of agricultural bio-inputs, they are widely applied in soybean, and increasingly accepted in crops such as corn and wheat with biological products mainly based on *Bradyrhizobium* spp., *Pseudomonas* spp., *Bacillus* spp. and *Azospirillum* spp. [7,29]. However, few studies describing PGPB’s effects on horticultural crops, particularly on lettuce, have been reported in Argentina, despite their importance. These studies showed the palliative effect that the use of *Azospirillum* spp. had on abiotic stresses [30,31,32] since better post-transplant behavior was found in lettuce plants from inoculated seeds subjected to salt stress. Also, a favorable response to co-inoculation with three strains of *A. argentinense* [33] and with species of the genus *Pseudomonas* [34] has been reported. However, the persistence of inoculated PGPB within the lettuce environment has rarely been described [23,35].

The aim of the present work was therefore to evaluate four selected bacterial strains of the genera *Azospirillum*, *Gluconacetobacter*, *Pseudomonas* and *Bacillus* with known and interesting activities and effects as PGPB (nitrogen fixation, phytohormones production, phosphate solubilization and biocontrol activity) on different plants [36,37,38,39] as an experimental multispecies bio-input to enhance the performance of lettuce var. *sagess*. Therefore, the compatibility of strains in experimental aqueous formulations as well as in inoculated plant-growth substrates was evaluated. Root colonization and plant-growth-promoting effects were evaluated in seedlings and in the productive stage to test different strategies of co-inoculation.

## 2. Results

### 2.1. Tracking of Bacterial Strains

Bacterial strains *A. argentinense* Az39, *B.* sp. Dm-B10, *G. diazotrophicus* PAL-5 and *P. protegens* Pf-5 in single formulation (hereafter called Az, Ba, Gl and Ps bio-input, respectively) and multispecies formulation (hereafter called Mz bio-input) were time-tracked by colony-counting in aqueous formulations (Figure 1(A.1)) and inoculated substrate (Figure 1(A.2)) as well as on the roots of lettuce seedlings grown in two plant-growth systems (Figure 1(B.1,B.2)).

#### 2.1.1. Effectiveness of Plate Colony-Counting Technique

The combination of specific culture medium and antibiotic resistance (see Section 4) allowed us to efficiently count bacterial colonies of each organism or the combination of them on the desired sample. Also, substrate sterilization proved to be efficiently performedsince bacterial growth was not observed in plates with antibiotic-containing media when serial dilutions of uninoculated substrate were plated.

#### 2.1.2. Bacterial Survival in Aqueous Formulation

All studied bacterial species survived in aqueous formulation with levels higher than 10^6^ colony forming units (CFU) mL^−1^ for at least 24 h either when alone or in Mz bio-input (Figure 2). The Mz bio-input contained similar orders of magnitude of viable cells of each PGPB compared to the single-strain bio-inputs determined at time “0”, and only Ba and Ps remained at similar levels after 24 h of storage (~10^6^ and 10^9^ CFU mL^−1^, respectively). In contrast, the populations of Az and Gl within the Mz bio-input were significantly reduced by approximately three orders of magnitude compared to single Az and Gl bio-inputs after they were kept for 24 h. The pH values of single-strain bio-inputs at the initial time were slightly acidic (Gl), close to neutrality (Az and Ps), and slightly alkaline (Ba), resulting in a Mz bio-input close to neutrality (pH 7.67 ± 0.06) (Table 1). The pH of all formulations within 24 h of storage remained the same.

#### 2.1.3. Bacterial Survival in Inoculated Substrate

All the bacterial species selected for the formulations survived in the substrate once they were inoculated at the required number of live cells, with population levels over 10^6^ CFU g^−1^ of substrate dry-weight (DW) after 24 h post-inoculation (Figure 2). Furthermore, all the strains either remained at the numbers applied or even increased their numbers within the substrate in a short storage period of 24 h. Both single- and co-inoculated substrates presented a slightly modified pH after the addition of the required bio-input (Table 1).

#### 2.1.4. Bacterial Root Colonization in Inoculated Lettuce Seedlings

No growth of microorganisms was evidenced when roots macerates of uninoculated plants were plated on antibiotic-containing specific culture media. Moreover, lettuce-plant colonization was not accompanied by inhibition of plant growth or any other macroscopically visible disease in any of the analysed plant-growth systems.

In both plant-growth systems (flasks and trays), single-strain and Mz inoculation led to root colonization of lettuce plants by all strains both 7 and 14 days post-inoculation (DPI) (Figure 3), since strains were re-isolated from these tissues. Single-strain inoculation in flasks with semi-solid agar led to bacterial population values that ranged from ~10^6^ to 10^8^ CFU g^−1^ of root fresh weight (FW). Inoculation with Mz bio-input under comparable conditions showed similar behaviour of the Az and Ps strains, but Ba and Gl presented a reduced number compared to single-strain inoculation (Figure 3). Single-strain- or Mz-inoculated lettuce seedlings grown in trays were colonized by all the bacterial strains, reaching similar population values both 7 and 14 DPI (~10^4^–10^6^ CFU g^−1^ of root FW).

### 2.2. Effect of the Experimental Bio-Inputs on Plant Growth of Lettuce

The plant-growth-promoting effect of aqueous formulations of PGPB was evaluated in both seedling and productive stages (Figure 1(B.2,B.3), respectively). Plant-growth inhibition did not occur, nor were macroscopically visible disease symptoms observed in plants inoculated with single or a mixture of bacteria in any of the growth systems analysed.

#### 2.2.1. Plant-Growth-Promoting Effect at Seedling Stage

After lettuce inoculation with single or a mixture of bacteria at sowing, both in controlled conditions (growth chamber) and in nursery greenhouse conditions (commercial plant nursery), seedlings inoculated with Mz bio-input presented a higher growth evidenced by the plant growth parameters measured compared to uninoculated ones and several of the single-inoculated treatments. Mz bio-input-inoculated plants showed a significantly higher leaf area than uninoculated ones 14 DPI (Figure 4 and Figure 5). Similar results were observed in seedlings inoculated only with Ba bio-input. Seedlings in these two treatments (Mz and Ba) also showed a higher rate of production of new leaves: they reached the development of two leaves in less than 10 DPI, whereas the rest of the single-strain-inoculated and uninoculated seedlings developed two leaves at least 14 DPI (Figure 4). Thirty DPI, seedlings of the aforementioned treatments also had the largest differences in leaf area compared to uninoculated plants. Moreover, seedlings inoculated with Az and Ps bio-input also showed a larger leaf area compared to uninoculated ones (Figure 4 and Figure 5) (photographs of single-inoculated lettuce seedlings grown in controlled conditions are shown in Appendix A). Regarding biomass accumulation, Mz bio-input addition at sowing significantly increased total seedling DW biomass by 44–60% compared to the uninoculated treatment (Figure 5 and Figure 6). This increase in the total DW biomass of the co-inoculated seedlings correlated, in most cases, with significant differences in both root- and aerial-part DW biomass (Figure 5 and Figure 6). However, lettuce seedlings inoculated with Ba and Gl bio-input were 20–44% heavier in terms of total dry biomass accumulation compared to uninoculated seedlings, showing significant differences in at least three of the four trials run at the seedling stage. The plant-growth-promoting effect of inoculation with Ba and Gl bio-input, and co-inoculation with Mz bio-input, was observed mainly in the aerial part of the seedlings, with statistically significant differences in almost all the seedling-stage assays (Figure 6). Regarding the plant-growth-promoting effect observed in the roots, Mz bio-input-inoculated seedlings showed the most reproducible results, with significant differences compared to the uninoculated treatment in the different seedling-stage trials, although the increases were lower than those observed for the aerial part (Figure 6).

#### 2.2.2. Plant-Growth-Promoting Effect at Productive Stage

The results observed with Mz bio-input showed the highest lettuce FW at commercial harvest (Figure 7). In two of the three assays carried out, co-inoculation at sowing and transplanting (Mz-ST treatment) resulted in a significant improvement of the commercial FW of the plants compared to uninoculated plants (26% and 42% higher, respectively). Moreover, Mz-ST statistically exceeds the effect of Az and Ps bio-inputs in plant FW. Although the plant FW of lettuce plants inoculated with Az, Ba, Gl and Ps bio-input was higher than those of uninoculated plants in all the assays performed, the differences were not significant, except for Ba and Gl.

## 3. Discussion

Combinations of beneficial microorganisms such as PGPB in multispecies bio-inputs are promising biotechnological tool to improve crop yield and quality, and a reliable and eco-friendly solution that might respond to the demands of production of vegetables [9]. However, there are not many bacterial multispecies bio-inputs available to be used in horticultural crops, and scientific studies are scarce, not only regarding the promoting effects of bacteria acting together, but also on their compatibility and persistence in the environment where they are applied [35]. Here, we selected PGPB species already described and applied as bacterial bio-inputs on many crops, but that have not been evaluated in multispecies bio-input formulations, particularly in horticultural crops. To our knowledge, this is the first report in which the behavior and action of the PGPB *A. argentinense* Az39, *G. diazotrophicus* PAL-5, *P. protegens* Pf-5 and *Bacillus* sp. Dm-B10 combined as an experimental multispecies bio-input were evaluated and contrasted with single-strain bio-inputs in a lettuce crop.

The PGPB species used to formulate the bio-input mixture must be compatible, which occurs whenever they have no growth-suppressive effect on each other during in vitro co-culture or during the plant root or rhizosphere colonization competition assay [9]. However, the composition of the culture medium might affect the outcome of the in vitro compatibility test, and microorganisms could even colonize different ecological niches, suggesting that in vitro incompatible strains may not interfere with each other’s growth on the root surface [9]. In our experiments, inoculated bacteria were closely time-tracked since their first contact and during the seedling stage until transplanting. In this way, validation of compatibility and persistence of the strains were performed in situ, in aqueous formulation and substrate, and in vivo, by testing root colonization. All the strains successfully remained at high numbers in aqueous formulations, though it is worth mentioning that *G. diazotrophicus* and *A. argentinense* populations in the Mz bio-input decreased by approximately two orders of magnitude (Figure 2). Both *P. protegens* Pf-5 and *Bacillus* spp. are known to produce a wide range of secondary products, which might be particularly inhibitory to other strains within the mixture [38,39]. Alternatively, such changes in microorganism populations might be due to the drastic change in pH between the single strains and the Mz bio-inputs (from 6–7 to 7.5). These changes observed in the bacterial populations in aqueous formulation were not observed in the substrate after Mz inoculation (Figure 2), and moreover, populations of each strain were higher than in the aqueous formulation, suggesting that multiple factors might positively modulate their behavior in this system. In addition, co-inoculated PGPB strains were able to efficiently colonize lettuce seedling roots grown in the two plant-growth systems, with lower populations in the substrate than in flasks (Figure 3), as normally occurs [21]. A reduction in viable cells of *A. brasilense* in roots of wheat seedlings was reported after co-inoculation with *Pseudomonas* spp. [34]. This inhibitory effect did not occur in our experiments, in line with the concept that interaction between *Pseudomonas* and *Azospirillum* taxa might be influenced by the species or even strains, as has been reported [40].

Regarding lettuce, growing seedlings in trays followed by transplanting is a production strategy widely used in horticultural activity. The first step in trays allows controlled seedling growth and uniformity in size, ensuring the production of quality seedlings. Seedlings must quickly attain a vigorous vegetative growth after transplanting, a crucial step for the correct implantation of the crop [41]. Therefore, it is evident that there is a need to obtain high-quality seedlings that might cope better with the post-transplant stresses. In line with this, beneficial effects of various bacterial genera, including some used in our experiments, have been reported when they are applied both pre- and post-transplant of lettuce plants [9,33]. Our results demonstrated that Mz bio-input always enhanced plant-growth compared to individually applied strains and the uninoculated control (Figure 4, Figure 5 and Figure 6). Mz bio-input was the only aqueous formulation that repeated higher and stable results in root, aerial part, and total DW, which occurred not only in controlled conditions but also in the nursery greenhouse, where biotic and abiotic factors have a greater impact on inoculant performance (Figure 6). This increase in root DW, linked to efficient root colonization by the four strains, most probably improved water uptake or management and absorption of nutrients, as well as physical support for the development of shoots and leaves. All this together probably led to an increase in shoot size which was shown by the larger biomass of the shoots of inoculated plants. In accordance with this, the Mz bio-input-treated seedlings had a higher growth rate (Figure 4A), which was shown by the higher number of leaves and the larger leaf area at the early stages (Figure 4B). Thus bigger and healthier seedlings were obtained in a shorter period of time, which also allowed earlier transplanting, which might shorten the cycle of production until harvest. Such a reduction in seedling growth time has already been reported when lettuce plants were co-inoculated with a mixture of *Azospirillum* strains [33]. Lettuce yield in productive greenhouse conditions behaved quite similarly to what was observed at the seedling stage: Mz bio-input was the only bio-input that consistently yielded larger lettuce plants in terms of FW compare to the uninoculated control (with at least one of the three inoculation strategies in each experimental station), which indicates that the farmers might obtain greater production and therefore more money. Plants inoculated with single-strain bio-input improved their FW compared to the uninoculated control, but at a lower level than plants inoculated with the Mz bacterial mixture (Figure 7). The most widely accepted hypothesis regarding PGPB’s mechanism of action postulates that a sum of events accounts for what happens when they are inoculated as a single bio-input, and certainly also when they are inoculated together. In this work the positive effect of Mz bio-input might be related to the ability to synthetize and release phytohormones, solubilize nutrients and/or fix atmospheric nitrogen among other activities that might also occur [36,37,38,39]. These multiple and complex mechanisms have been described in microbe–plant interactions, but it is difficult to identify any of them individually as supporting the changes in plant growth, especially in a mixture. Therefore, several simultaneous mechanisms could be working together to shape the outcome of the Mz strains’ interactions, with *Bacillus* sp. and *G. diazotrophicus* as possible main contributors to such an effect. Therefore, further studies will be necessary to understand the interactions among the four strains of the multispecies bio-input, evaluating other inoculation strategies and other crop conditions. Nevertheless, this work is one of the few studies showing the synergistic effects of a multispecies bacterial bio-input on lettuce plants in terms of plant-growth-promoting effects. The findings reported here provide relevant information on the establishment of the four selected strains to co-inoculate lettuce at the seedling stage, and also ensure that results found in the laboratory could be reproduced in nursery/productive greenhouse conditions. These contributions add to those already reported in relation to multispecies bacterial bio-input application to enhance the efficiency of the use of nutrients and to increase crop yields. In addition, our results contribute to encouraging studies regarding the formulation, characterization and use either at the nursery or in production farms of synthetic mixtures of organisms in order to provide high-quality, productive plants that allow farmers to improve their production under a sustainable production system.

## 4. Materials and Methods

### 4.1. Bacterial Strains and Maintenance

The bacterial strains used in this study were *Azospirillum argentinense* Az39 (syn. *Azospirillum brasiliense*) [42,43], *Pseudomonas protegens* Pf-5 (syn. *Pseudomonas fluorescens*) [44,45], both strains belonging to the Bacterial Culture Collection of the Instituto de Microbiología y Zoología Agrícola (IMYZA) of INTA-Castelar, Buenos Aires, Argentina; *Gluconacetobacter diazotrophicus* PAL-5 [46,47] kindly provided by Dr. Caballero-Mellado; and *Bacillus* sp. Dm-B10 from the Bacterial Culture Collection of the Centro de Investigaciones en Fitopatología (CIDEFI) [48]. Bacterial strains were grown in specific culture media (Table 2) and maintained at 4°C with monthly subcultures. The same specific culture media containing 20% glycerol [*v*/*v*] at –80°C was used to store bacterial strains. The outstanding traits of each strain as PGPB are summarized in Table 2.

### 4.2. Bacterial Cultures and Formulation of Experimental Bio-Inputs

Bacterial strains were individually grown in flasks containing liquid medium (Table 1) on a rotatory shaker at 150 rpm and 30 °C for 24 h (*Bacillus* sp. and *P. protegens*) or 48 h (*A. argentinense* and *G. diazotrophicus*). Then, these cultures were used as inoculum (10% of final volume) for growing each strain in the same conditions for 48 h to provide the inocula for the experimental bacterial bio-inputs. The concentration of colony-forming units (CFU) per ml and final pH of bacterial inocula are detailed in Table 1. Aqueous formulations of experimental single bio-inputs were obtained by making a 1:5 dilution of each complete bacterial inoculum using sterile distilled water (pH 7.0). The aqueous formulation of the Mz bio-input was formulated by mixing equal volumes of each bacterial inoculum (Table 1) with sterile distilled water (pH 7.0), obtaining a 1:5 final dilution of each strain.

### 4.3. Bacterial Survival Assay

The natural Antibiotic resistance of strains [51,52,53] (Table 1) was used to evaluate bacterial survival in Az, Ba, Gl, Ps and Mz bio-inputs and in plant-growth substrate using a plate colony-counting technique (Figure 1(A.1,A.2), respectively).

#### 4.3.1. Bacterial Survival in Aqueous Formulations

Bacterial survival was assayed by taking 10 mL of each bio-input 0 and 24 h after formulation and maintenance in the dark at 18–20 °C. The appropriate dilutions of bacterial suspensions (from 10^−2^ to 10^−7^) were plated in each specific antibiotic-containing culture medium and CFU were counted after 48–72 h of incubation at 30 °C. At the same time, the pH of each sample was tested. Two independent assays with three replicates each were performed and the means of the logarithm of CFU mL^−1^ of each experimental bio-input were plotted.

#### 4.3.2. Bacterial Survival in Inoculated Substrate

Bacterial survival was assayed in a 1:1 mixture of organic compost and amended soil (pH 6.4 and 6.5, respectively, Rincón Verde–Biofertyl SRL) filled in cells of 10 mL seedling trays. The substrate was autoclaved at 121 °C for 40 min to decrease the microbe population. Experimental bio-inputs were added at a ratio of 1 mL per cell. Distilled sterile water was used instead of the experimental bio-inputs as control (U treatment). A modified protocol described by [54] was used to isolate bacteria from substrate samples. In triplicate and separately, 5 g of uninoculated and inoculated substrate were taken from the cells and shaken for 40 min at 150 rpm in sterile flasks with 45 mL of phosphate buffer solution (PBS) (8.00 g L^−1^ NaCl; 0.20 g L^−1^ KCl; 1.44 g L^−1^ Na_2_HPO_4_; 0.24 g L^−1^ KH_2_PO_4_) with tween 80 (0.01% *v*/*v*). The appropriate serial dilutions (from 10^−2^ to 10^−7^) were plated in the specific antibiotic-containing culture media and CFU were counted after 48–72 h of incubation at 30 °C. This procedure was repeated 0 and 24 h after substrate inoculation and the pH (substrate:water ratio of 1:2.5) of each sample was also determined using a potentiometric method. Two independent assays with three replicates each were performed and the means of the logarithm of CFU g^−1^ of substrate FW were plotted.

### 4.4. Plant Growth Conditions and Inoculation

Lettuce (*Lactuca sativa* L.) var. *sagess* BRP5938 (Vilmorin) seeds, kindly provided by Baby Plant S.A (La Plata, Argentina), were used in all plant-inoculation experiments carried out at the seedling and productive stage as described below.

#### 4.4.1. Lettuce Plants at Seedling Stage

Lettuce plants were grown at the seedling stage in two plant-growth systems to evaluate root colonization and/or plant-growth-promotion effects after bacterial inoculation (Figure 1(B.1,B.2)).

Plant-growth system 1: flasks with sterile semisolid agar-containing medium and seeds, disinfected to diminish the presence of indigenous microorganisms, were used in controlled conditions (plant-growth chamber at 26 °C with photoperiod light/dark cycle of 16 and 8 h, respectively), a suitable environment to study plant-bacteria interactions. Lettuce seeds were surface-disinfected with 20% *v*/*v* sodium hypochlorite (46 g L^−1^ NaClO) for 10 min and washed three times with PBS [21,55]. Disinfected seeds were placed between wet sterile filter papers in the dark and incubated for ~48 h at 28 °C for germination. Sprouted seeds were immersed in each experimental bio-input (Az, Ba, Gl, Ps and Mz bio-input) for 20 min without agitation and placed on flasks containing 100 mL of sterile semisolid Fåhraeus agar-containing medium (0.5% *w*/*v*) at pH 6.00 [56]. Distilled sterile water was used instead of experimental bio-inputs as the uninoculated control (U treatment). Seedlings were grown in controlled conditions throughout the assay. Two independent assays were performed using a completely randomized design with six treatments and three biological replicates each: uninoculated, single-inoculated with Az, Ba, Gl or Ps bio-inputs, and co-inoculated with Mz bio-input (U, Az, Ba, Gl, Ps and Mz treatments, respectively). Samples were taken on different days to evaluate root colonization as described below (Figure 1(B.1)).Plant-growth system 2: trays with sterile/nonsterile substrate and disinfected/nondisinfected seeds were used to study plant–bacteria interactions in controlled conditions (growth chamber) and in nursery greenhouse conditions (a commercial plant nursery) (Figure 1(B.2)).
-*Lettuce seedlings grown in controlled conditions*: surface-disinfected seeds were placed in seedling trays filled with an autoclaved 1:1 mixture of organic compost and amended soil (pH 6.4 and 6.5, respectively; Rincón Verde, Biofertyl–SRL) as substrate. Experimental Az, Ba, Gl, Ps and Mz bio-inputs were added on seeds (immediately after formulating the bio-inputs) at a ratio of 1 mL per cell (10 mL of substrate), after which seeds were covered with a thin layer of substrate. Sterile distilled water was used instead of experimental bio-inputs as the uninoculated control (U treatment). Seedling trays were grown in controlled conditions throughout the test. Plants were kept with irrigation on demand and modified Hoagland’s solution with KNO_3_ as the nitrogen source (0.14 g L^−1^ KH_2_PO_4_; 0.50 g L^−1^ KNO_3_; 1.18 g L^−1^ Ca_2_(NO_3_)_2_·4H_2_O; 0.49 g L^−1^ MgSO_4_·7H_2_O; 2.86 mg L^−1^ H_3_BO_3_; 1.81 mg L^−1^ MnCl·4H_2_O; 0.22 mg L^−1^ ZnSO_4_·7H_2_O; 0.08 mg L^−1^ CuSO_4_·5H_2_O; 2 mg L^−1^ Na_2_Mo_4_·2H_2_O; 3 mg L^−1^ FeSO_4_·7H_2_O; 3.73 mg L^−1^ EDTANa_2_) [57] which was added once a week after 14 DPI. Two independent assays were performed using a completely randomized design with six treatments and three biological replicates each: uninoculated, inoculated with Az, Ba, Gl or Ps bio-inputs, and inoculated with Mz bio-input (U, Az, Ba, Gl, Ps and Mz treatments, respectively). Samples were taken on different days to evaluate root colonization, biomass accumulation and leaf area as described below.-*Lettuce seedlings grown in nursery greenhouse conditions*: Following the workflow usually carried out by the commercial plant nursery, Baby Plant S.A. (−34.980211, −58.031761, La Plata, Argentina), for cropping lettuce seedlings, nondisinfected lettuce seeds were sowed with a pneumatic sowing machine in seedling trays (volume of 10 mL per cell) with nonsterile peat-based substrate (Klasmann Ts1). Inoculation was performed as previously described for seedlings grown in trays in controlled conditions. Seedling trays were kept in nursery greenhouse conditions and irrigated on demand throughout the test. Two independent assays (October 2018 and September 2019) were carried out using a completely randomized design with six treatments and three biological replicates each: uninoculated, inoculated with Az, Ba, Gl or Ps bio-inputs, and inoculated with Mz bio-input (U, Az, Ba, Gl, Ps and Mz treatments, respectively). Biomass accumulation was evaluated as described below when seedlings were ready for transplanting (30–35 DPI and 4 to 6 developed leaves).

#### 4.4.2. Lettuce Plants at Productive Stage

Lettuce plants were grown in productive greenhouse conditions in two experimental sites to evaluate plant-growth-promotion effects after bacterial inoculation (Figure 1(B.3)). Seedlings from commercial plant nursery assays were transplanted into the greenhouses of two experimental stations located in the CHP. (I) Estación Experimental Julio Hirschhorn (hereafter called EE JH, −34.985851, −57.997866, Facultad de Ciencias Agrarias y Forestales, Universidad Nacional de La Plata); and (II) Estación Experimental Gorina (hereafter called EE Gorina, −34.914597, −58.038621, Ministerio de Desarrollo Agrario, Provincia de Buenos Aires). Two trials were conducted at EE JH in November 2018 and October 2019, and one trial was conducted at EE Gorina in October 2019. The treatments were performed as follows: 30 DPI seedlings inoculated at sowing were re-inoculated before transplanting, giving rise to the Az-ST, Ba-ST, Gl-ST, Ps-ST and Mz-ST treatments (ST: inoculation at sowing and before transplanting); 30 DPI seedlings inoculated with Mz bio-input at sowing were not re-inoculated before transplanting, giving rise to the Mz-S treatment (S: inoculation only at sowing). Furthermore, 30 DPI uninoculated seedlings were divided into two treatments: (i) seedlings inoculated with Mz bio-input before transplanting, giving rise to the Mz-T treatment (T: inoculation only before transplanting) and (ii) seedlings without inoculation before transplanting, giving rise to the uninoculated control treatment (U). In all cases, inoculation was performed by overnight immersion of trays in the corresponded experimental bio-input, at a rate of 2 mL of bio-input per seedling cell (10 mL substrate). Sterile distilled water (pH 7) was used instead of bio-input for the uninoculated treatment. After inoculation, seedlings were transplanted into furrows with a plant density of 16 plants per m^2^ and watered using drip irrigation on demand until commercial harvest (45/60-days post-transplant, DPT). The tests were carried out without fertilization, and chemical properties of each soil are shown in Table 3. Three independent assays were performed using a randomized complete block design with eight treatments (Az-ST, Ba-ST, Gl-ST, Ps-ST, Mz-S, Mz-T, Mz-ST and U) with four biological replicates each.

### 4.5. Plant-Root Colonization Assessment

Lettuce seedlings grown both in flasks and in trays in controlled conditions were removed 7 and 14 DPI to evaluate root colonization by strains using a plate colony-counting technique. In triplicate for all samples, root systems were separated from aerial parts and vigorously washed with sterile PBS. Roots were macerated using a mortar and pestle, and homogenized macerates were resuspended in 1 mL of sterile PBS, vortexed, 10-fold serially diluted, plated in each antibiotic-containing specific medium (Table 1) and incubated at 30 °C for 48–72 h. CFU were counted to quantify the root population as described by [60]. Results were expressed as the mean logarithm of the CFU g^–1^ of root FW per treatment for each assay.

### 4.6. Plant-Growth Promotion Assessment

The appearance of new leaves (considered when a new leaf is observed in at least 50% of seedlings, expressed in DPI), leaf area (14 and 30 DPI) and dry biomass accumulation in root and aerial parts 30 DPI were measured in seedlings grown in controlled conditions. Image J software v1.53e [61] was used to determine the leaf area of seedlings (first and second leaf). The dry biomass accumulation in root and aerial parts at 30 DPI was measured in seedlings grown in nursery greenhouse conditions. Dry biomass accumulation was measured in quadruplicate for each treatment by drying the roots and aerial parts in an oven at 60 °C to a constant weight. Plant-growth promotion in the productive stage was evaluated by measuring plant fresh weight at commercial harvest (45–60 DPT).

### 4.7. Statistical Data Analysis

Infostat software [62] was used to perform analysis of variance (ANOVA) and Tukey’s test on the data. Bifactorial ANOVA was performed to analyse the CFU ml^−1^ or g^−1^ of each strain (considered as individual variables) for survival and colonization assays. The two factors considered were formulation (with two levels: single and multispecies) and time (with two levels: 0 and 24 h or 7 and 14 DPI). ANOVA was used to evaluate differences in root, aerial part and total DW between the six treatments (Az, Ba, Gl, Ps, Mz and U) in seedling-stage trials; data were analysed individually for each trial and altogether considering each trial as a block. ANOVA was performed to analyse the leaf area data of the six treatments (Az, Ba, Gl, Ps, Mz and U), considering the first and second leaf as a block. ANOVA were performed to analyse the eight treatments in productive-stage trials (Mz-S, Mz-T, Mz-ST, Az, Ba, Gl, Ps and U); data were analysed individually for each trial and altogether considering each trial as a block. When necessary, comparisons of means were conducted using Tukey’s post-hoc test and *p* ≤ 0.05 were considered statistically significant. In all parametric statistical tests applied, the assumptions of normality and homoscedasticity of residuals were confirmed (95% confidence) with the modified Shapiro–Wilks and Cochran tests, respectively.

## 5. Conclusions

Several interesting conclusions can be drawn from our results. First of all, the synthetic mixtures of organisms can be successfully used to promote vegetable production provided that several characteristics of the formulation are considered. Regarding this, PGPB *A. argentinense* Az39, *G. diazotrophicus* PAL-5, *P. protegens* Pf-5 and *Bacillus* sp. Dm-B10 can be used together to promote plant growth. These PGPB should be packaged so that they are mixed at the moment they are applied, since when kept in liquid media their interaction led to changes in their relative numbers, probably affecting the effect on plant-growth promotion.

The environment plays a key role in organisms’ ability to survive and grow on quite similar bases since plants were found to be colonized by the four species of bacteria applied, unlike what was observed in liquid media. Based on this, plant management may also have a large impact on PGPB colonization and survival and therefore on growth promotion.

Co-inoculation did not affect PGPB colonization ability and provided plants with additional capacities, probably supplied by each of the PGPB inoculated, which lead to healthier and more productive plants.

Therefore, synthetic mixtures of bacteria appear to be a feasible tool to promote plant growth and keep plants healthy; however, future work should be aimed at analysing the interactions that occur within organisms at the plant level as well as on the formulation media.

## Figures and Tables

**Figure 1 plants-12-00736-f001:**
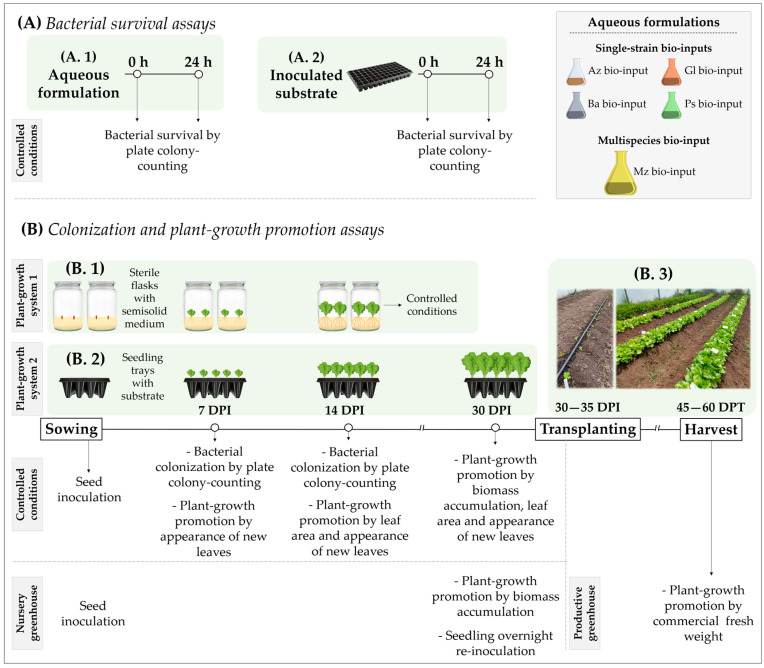
Graphical abstract of tracking and plant-growth–promotion assays of bacterial strains. Seedling time is expressed in days post-inoculation (DPI) and days post-transplanting (DPT).

**Figure 2 plants-12-00736-f002:**
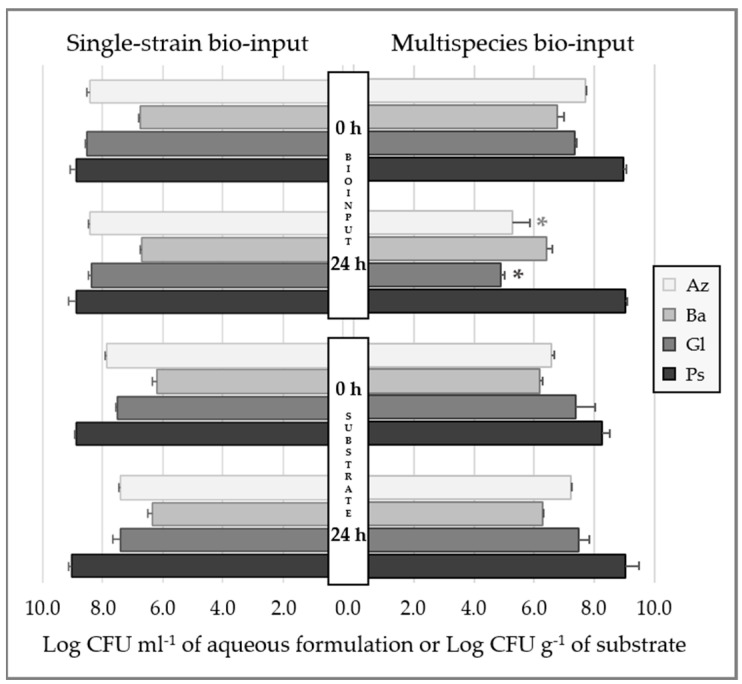
Bacterial populations of each strain in aqueous formulation and in inoculated substrate at time “0” and after 24 h of storage post-formulation or post-inoculation, respectively. In the figure, average of log CFU per ml of bio-input and average of log CFU per g of substrate (and standard deviation of data) of each strain are plotted. Az: *A. argentinense* Az39; Ba: *Bacillus* sp. Dm-B10; Gl: *G. diazotrophicus* PAL5; Ps: *P. protegens* Pf5. An asterisk above the bar indicates significant differences between means.

**Figure 3 plants-12-00736-f003:**
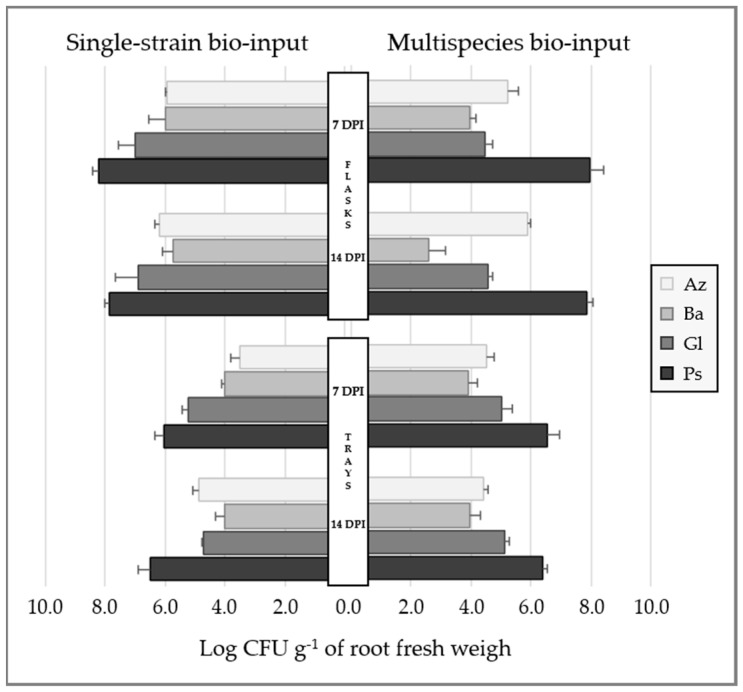
Root bacterial populations of each strain in lettuce seedlings grown in different plant-growth systems (flasks and trays) 7 and 14 days post-inoculation (DPI). In the figure, the average of log CFU per g of root fresh weight (and standard deviation of data) of each strain 7 and 14 DPI of seedlings grown in flasks and trays are plotted. Az: *A. argentinense* Az39; Ba: *Bacillus* sp. Dm-B10; Gl: *G. diazotrophicus* PAL5; Ps: *P. protegens* Pf5.

**Figure 4 plants-12-00736-f004:**
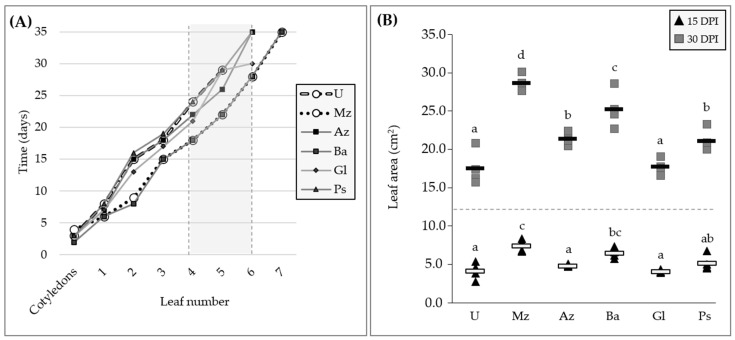
(**A**) Variation in number of leaves as a function of time in days post-inoculation and (**B**) leaf area 14 and 30 days post-inoculation (DPI) with each bio-input in lettuce seedlings. In the figure, average of days after which new leaves appear (**A**) and leaf area (**B**) are plotted. Different letters in the statistical data analysis (on the right) indicate significant differences (*p* ≤ 0.05) according to Tukey’s test. Mz: multispecies; Az: *A. argentinense* Az39; Ba: *Bacillus* sp. Dm-B10; Gl: *G. diazotrophicus* PAL5; Ps: *P. protegens* Pf5; U: uninoculated.

**Figure 5 plants-12-00736-f005:**
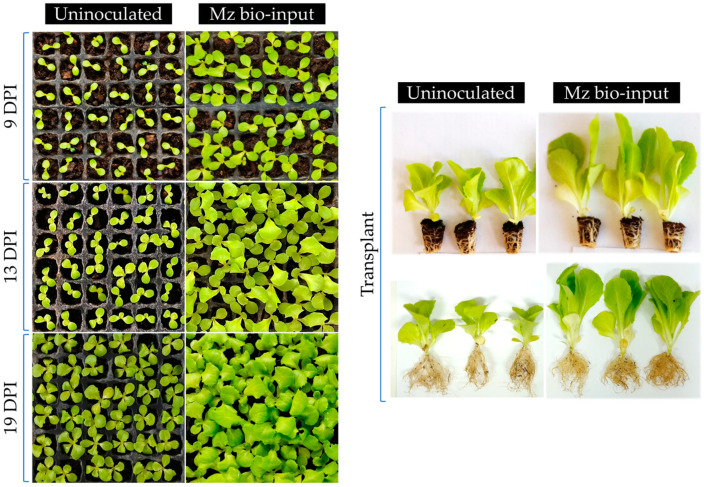
Photographs of uninoculated/inoculated lettuce seedlings grown in controlled conditions after several days post-inoculation (DPI) with Mz bio-input.

**Figure 6 plants-12-00736-f006:**
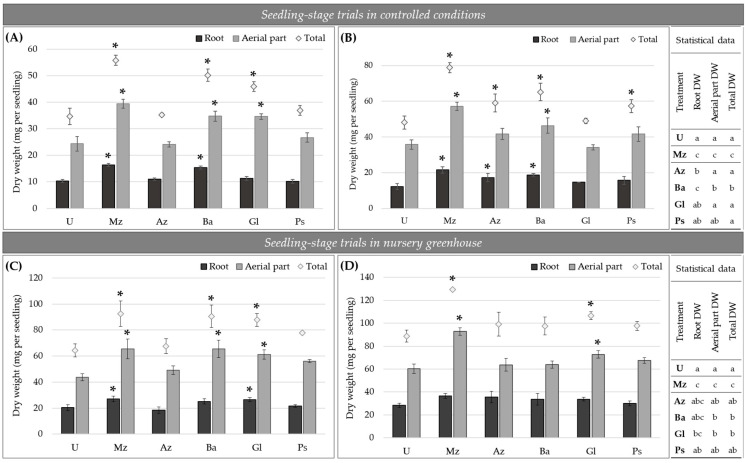
Root, aerial part, and total dry-weight (DW) biomass of lettuce seedlings grown in controlled conditions ((**A**,**B**), independent assays) and in nursery greenhouse ((**C**,**D**), independent assays) 30 days post-inoculation (DPI). In the figure, means of root DW (dark-gray bar), aerial part DW (light-gray bar), and total DW biomass (light-gray diamonds) (and standard deviation of data) of seedlings 30 DPI are plotted. Asterisks above the bar indicate significant differences with respect to the U treatment in each individual trial, at *p* ≤ 0.05 according to Tukey’s test. Different letters in the statistical data (on the right) indicate significant differences (*p* ≤ 0.05) according to Tukey’s test. Mz: multispecies; Az: *A. argentinense* Az39; Ba: *Bacillus* sp. Dm-B10; Gl: *G. diazotrophicus* PAL5; Ps: *P. protegens* Pf5; U: uninoculated.

**Figure 7 plants-12-00736-f007:**
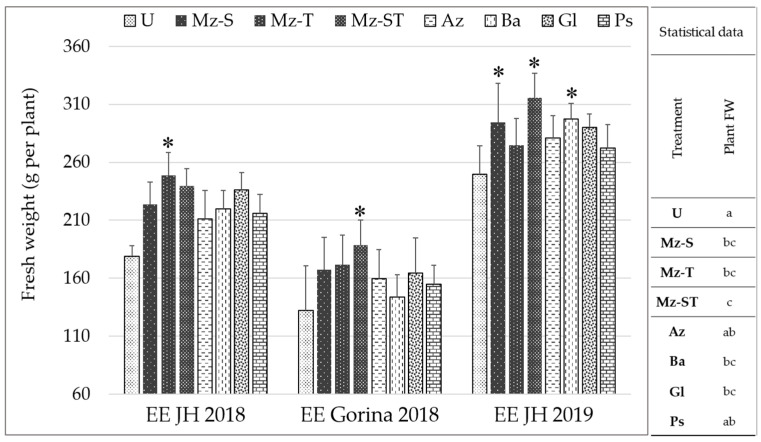
Lettuce yield of plants grown in productive greenhouse conditions. In the figure, average of lettuce plants fresh weight (FW) at commercial harvest of plants in each experimental station are plotted. Asterisks above the bar indicate significant differences with respect to the U treatment in each individual trial at *p* ≤ 0.05 according to Tukey’s test. Different letters in the statistical data (on the right) indicate significant differences (*p* ≤ 0.05) according to Tukey’s test. Mz-S: multispecies at sowing; Mz-T: multispecies before transplanting; Mz-ST: multispecies at sowing and before transplanting; Az: *A. argentinense* Az39; Ba: *Bacillus* sp. Dm-B10; Gl: *G. diazotrophicus* PAL5; Ps: *P. protegens* Pf5; U: uninoculated.

**Table 1 plants-12-00736-t001:** pH values of aqueous formulations and substrate at time “0” and after 24 h of storage post-formulation or post-inoculation, respectively.

System	Time	Single-Strain Bio-Input ^1^				Mz Bio-Input ^2^	Uninoculated
		Az	Ba	Gl	Ps		
Aqueous formulation	0 h	7.01	8.30	6.16	7.58	7.67	--
	24 h	6.91	8.22	6.26	7.54	7.48	
Substrate	0 h	6.45	6.47	6.50	6.50	6.74	6.27
	24 h	6.75	6.73	6.60	6.40	6.76	6.26

^1^ Az: *A. argentinense* Az39; Ba: *Bacillus* sp. Dm-B10; Gl: *G. diazotrophicus* PAL5; Ps: *P. protegens* Pf5. ^2^ multispecies.

**Table 2 plants-12-00736-t002:** PGPB traits, antibiotic resistance, culture media final pH, and concentrations of bacterial cultures.

Strain	Plant-Growth Promoting Traits [36,37,38,39]	Antibiotic Resistance ^1^	Culture Media ^2^	pH Initial/Final	Concentration ^3^
*Azospirillum argentinense* Az39	-Phytohormone production -Nitrogen fixation	Kanamycin	CR	6.8/7.0	~10^9^
*Bacillus* sp. Dm-B10	-Biocontrol activity	Rifampicin	LB	7.0/8.3	~10^8^
*Gluconacetobacter diazotrophicus* PAL5	-Nitrogen fixation -Phosphate solubilization	Nalidixin	LGI	6.0/6.16	~10^9^
*Pseudomonas protegens* Pf-5	-Phosphate solubilization -Biocontrol activity	Streptomycin	LB	7.0/7.6	~10^9^

^1^ Natural antibiotic resistance. Kanamycin 100 μg mL^−1^; rifampicin 100 μg mL^−1^; nalidixin 15 μg mL^−1^; streptomycin 400 μg mL^−1^. ^2^ CR: Congo Red medium [42], LB: Luria Bertani medium [49], LGI: liquid glucose Ivo medium [50]. ^3^ CFU mL^−1^ of bacterial culture, determined using plate colony-counting technique.

**Table 3 plants-12-00736-t003:** Chemical analysis of soil from Julio Hirschhorn (EE JH) and Gorina (EE Gorina) experimental stations.

Analysis ^1^	Units	EE JH	EE Gorina
pH		6.81	7.49
EC	[dS.m^−1^]	1.28	1.25
C	[%]	2.54	1.39
OM	[%]	4.38	2.40
Nt	[%]	0.242	0.134
C/N		10	10

^1^ pH: potentiometric determination, soil:water ratio 1:2.5. EC: electric conductivity. C: organic carbon [58]. OM: organic matter. Nt: total nitrogen [59]. C/N: carbon nitrogen relation.

## Data Availability

The data that support the findings of this study are available upon request from the corresponding author.

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
