# Peer review of "Multispecies Bacterial Bio-Input: Tracking and Plant-Growth-Promoting Effect on Lettuce var. sagess"

_plants, 2023, doi:10.3390/plants12040736_

Round 1

Reviewer 1 Report

This study shows the synergistic effects of a bacterial multispecies bio-input on lettuce plants in terms of plant-growth–promoting effects.

I found the paper to be overall very well written and organized. The statistical and graphic evidence shown is very clear about the application of a multispecies bacterial bioinput, compared to inoculation with a single species. This sheds light on a controversial subject, on which there is not much information for horticultural crops. The results are very well presented and discussed, comparing with other works. Also, the paper presents an updated review of the cited bibliographical references, all pertinent to the subject of study.

Despite the good scientific quality of the paper, I have some comments about it, indicated below:

- It is not clear if the bacteria were washed for the preparation of the inoculum, since the components of the culture media could interfere with the results of the experiments.

- It would be convenient to briefly include the main plant growth-promoting characteristics of each bacterium used in this study. This would support the last part of the discussion of the results.

Line 266: Change A. brasilense for A. argentinense.

Figure 6. Indicate what is shown in each panel of the figure (for example, A, B, C, D), since the only thing that varies is the dry weight scale, and of course, the results.

Author Response

Dear reviewer,

We are grateful for your contributions to improve the manuscript and we have endeavoured to incorporate all the suggestions. We have also re-edited the manuscript by an English speaker. These changes can be checked in the revised version with change control.

Responses of your comments:

Comment #1 -It is not clear if the bacteria were washed for the preparation of the inoculum, since the components of the culture media could interfere with the results of the experiments.

Answer: We understand this reviewer's comment and added “complete” in the manuscript. See Line 418.

For the bio-input formulations we used the complete inoculum of each strain, without centrifugation, as commercial bio-inputs are usually applied. We consider that the components of the culture media remain in very low concentration in the bacterial inocula since they decrease considerably due to their consumption by the microorganisms. Due to this and also to the dilution made when formulating the bio-inputs, we consider that their contribution can be considered negligible.

Comment #2 -It would be convenient to briefly include the main plant growth-promoting characteristics of each bacterium used in this study. This would support the last part of the discussion of the results.

Answer: We followed this reviewer´s suggestion. The outstanding traits of each strain as PGPB were summarized in introduction (Line 118-119) and in Table 2 in materials and methods.

Comment #3 Line 266: Change A. brasilense for A. argentinense.

Answer: It was done, see Line 309.

Comment #4 -Figure 6. Indicate what is shown in each panel of the figure (for example, A, B, C, D), since the only thing that varies is the dry weight scale, and of course, the results.

Answer: We followed this reviewer´s suggestion and added letters in each panel of Figure 6 and indicated what is shown.

Reviewer 2 Report

This manuscript reports a study focussed on the comparison of plant-growth promoting effect single-inoculation and bacterial-multispecies inoculation.

The manuscript is well written and clearly presented. The conclusions are sound and well supported by the obtained results.

Given this the manuscript can be published in its present form.

Author Response

Dear reviewer,

We are grateful for your contributions to improve the manuscript and we have endeavoured to incorporate all the suggestions. We have also re-edited the manuscript by an English speaker. These changes can be checked in the revised version with change control.

Reviewer 3 Report

Dear Authors

The article plants-2158230 is very interesting and well-written. The idea of ​​the experiment is respectable and the experiments seem well organized.

It shows us the importance and significance of synergism of a bacterial multispecies bio-input on an important worldwide crop, the lettuce.

Authors tested four selected bacterial strains with plant-growth–promoting effects on different plants to enhance the performance of lettuce and they experimented with the compatibility issue of these strains.

·         I suggest that the authors add to the discussion a few sentences about cases of beneficial bacteria, as single or as mixtures, in hydroponic lettuce growing conditions, which is something very common in recent years and of great interest to many researchers.

·         Also, authors can rewrite the “Conclusions” to make the results look more attractive

·         Figure 1 in my copy is not at all clear. It needs to be sharper.

Author Response

Dear reviewer,

We are grateful for your contributions to improve the manuscript and we have endeavoured to incorporate all the suggestions. We have also re-edited the manuscript by an English speaker. These changes can be checked in the revised version with change control.

Responses of your comments:

Comment #1 -I suggest that the authors add to the discussion a few sentences about cases of beneficial bacteria, as single or as mixtures, in hydroponic lettuce growing conditions, which is something very common in recent years and of great interest to many researchers.

Answer: We strive to add some information about applied of PGPB in hydroponic lettuce growth but we did not find the right place to add it.

Comment #2 -Also, authors can rewrite the “Conclusions” to make the results look more attractive

Answer: We agree with this reviewer's comment and we strive to improve the discussion and conclusions.

Comment #3 -Figure 1 in my copy is not at all clear. It needs to be sharper.

Answer: We agree with this reviewer's comment and changed the font size and some other things to make Figure 1 more understandable.

Reviewer 4 Report

I recommend the publication of the article because scientific experimentation on the use of products with multi-species microbial consortia to increase the growth and resistance of plants to diseases is of particular interest.

The aim and objectives of the article have been stated and are very interesting. The use of multi-species microbial consortia is an important topic especially for the reduction of synthetic products in agriculture and for increasing soil and plant colonisation. The work done is certainly of international interest and the format applied is certainly suitable for a research paper. The work done is original, of particular interest and can certainly stimulate research on this topic. The length of the article is appropriate for the journal and the graphs and tables are clear and easy to understand. The conclusion summarises the aims of the work and future prospects.

Author Response

(The authors gave the same response as above.)
